# Inertial Sensor Estimation of Initial and Terminal Contact during In-Field Running

**DOI:** 10.3390/s22134812

**Published:** 2022-06-25

**Authors:** Yue Yang, Li Wang, Steven Su, Mark Watsford, Lauren Marie Wood, Rob Duffield

**Affiliations:** 1Faculty of Engineering and IT, University of Technology Sydney, 81 Broadway, Ultimo, NSW 2007, Australia; yue.yang-4@student.uts.edu.au (Y.Y.); li.wang-9@student.uts.edu.au (L.W.); steven.su@uts.edu.au (S.S.); 2School of Sport, Exercise and Rehabilitation, Faculty of Health, University of Technology Sydney, Moore Park, NSW 2007, Australia; mark.watsford@uts.edu.au; 3Graduate School of Biomedical Engineering, Samuels Building (F25) Library Rd, UNSW, Kensington, NSW 2052, Australia; lauren.wood1@unsw.edu.au

**Keywords:** gait analysis, acceleration, angular velocity, inertial measurement device

## Abstract

Given the popularity of running-based sports and the rapid development of Micro-electromechanical systems (MEMS), portable wireless sensors can provide in-field monitoring and analysis of running gait parameters during exercise. This paper proposed an intelligent analysis system from wireless micro–Inertial Measurement Unit (IMU) data to estimate contact time (CT) and flight time (FT) during running based on gyroscope and accelerometer sensors in a single location (ankle). Furthermore, a pre-processing system that detected the running period was introduced to analyse and enhance CT and FT detection accuracy and reduce noise. Results showed pre-processing successfully detected the designated running periods to remove noise of non-running periods. Furthermore, accelerometer and gyroscope algorithms showed good consistency within 95% confidence interval, and average absolute error of 31.53 ms and 24.77 ms, respectively. In turn, the combined system obtained a consistency of 84–100% agreement within tolerance values of 50 ms and 30 ms, respectively. Interestingly, both accuracy and consistency showed a decreasing trend as speed increased (36% at high-speed fore-foot strike). Successful CT and FT detection and output validation with consistency checking algorithms make in-field measurement of running gait possible using ankle-worn IMU sensors. Accordingly, accurate IMU-based gait analysis from gyroscope and accelerometer information can inform future research on in-field gait analysis.

## 1. Introduction

During running, the detection of two essential events is required to understand the basic step-by-step gait outcomes of running mechanics, including heel strike or initial contact (IC) and toe-off or terminal contact (TC) [1]. Determination of these events allow the calculation of temporal parameters such as foot contact time, flight phase duration, and swing phase duration [2]. IC is defined as the instance when one foot is initially touching or landing on the surface, while TC represents the termination of the pushing phase or when the foot finishes contact with the surface [3]. These two temporal parameters are intrinsically related and inform other key performance outcomes related to running, such as running economy [4], speed of running performance [5] and running related injuries and risks [6]. Hence, the potential application of in-field monitoring of these factors by wearable microtechnology in all running-based sports is substantial and methods to determine IC and TC are of paramount importance.

Gait research in many laboratories requires data collection tools that are difficult to use in field-based settings or in most team-sports where training is undertaken concurrently by large groups. For example, Wang et al. [7] reviewed the current vison-based motion tracking systems and found that these approaches required an optimal range of object-to-camera distance, which is unlikely in field settings given the unpredictable movement patterns. This was further supported by Norris et al. [8] who undertook a systematic review of current gait analysis methods, suggesting that gait analysis using appropriately equipped laboratories with high-speed cameras or force platforms resulted in limitations due to the lack of appropriate simulation of outdoor or in-field running. Hence, there is a need to develop methods with easily accessible and portable tools, allowing gait analysis to be undertaken in field-based settings of larger groups that facilitate ecologically valid scenarios [9]. Accordingly, the application of inertial measurement units (IMU), which include gyroscopes and accelerometers, to the lower limbs may assist with enhancing the quality of gait assessment in field-based contexts. Such devices can capture the resistance in linear or angular motions [10], have low manufacturing cost, high sampling frequencies, reduced size and weight, and are able to monitor kinematics throughout an extended period [4]. These advantages can make IMU a desirable system to investigate running-based parameters. In support, a recent review [11] examined the current applications and research of wearable inertial sensor across different sports, suggesting these types of sensors suited in-field analysis of running gait analysis.

Current approaches differ for the collection of IC and TC using IMU, especially based on the IMU placement and use of collected data [12]. Mo and Chow [13] conducted a review of current gait research, reporting that the major locations of placing IMU for gait analysis were the pelvis, shank, and foot. They concluded that particular sites for IMU placement required different data analysis methods to detect IC and TC, which could result in diverse measurement outcomes. Considering signal detection, Clark et al. [14] investigated tibial acceleration around the knee joint, particularly in the mediolateral plane, by examining varus/valgus knee movement throughout running. Furthermore, Muro-De-La-Herran et al. [15] studied gait movement using accelerometers and established that unit positioning such as shank, limb, and back was vital in delivering accurate analysis requiring different analysis algorithms. In terms of using gyroscopes, Bergamini et al. [16] analysed the difference between elite and sub-elite runners using an IMU containing a tri-axial gyroscope located at the lumbar spine region. They discovered no consistent signal from the acceleration data and designed an algorithm based on the angular velocity data which acquired an average error of less than 0.025 s. Ankle-based gait analysis has also been studied, as Qiu et al. [17] designed a smart system which detected the human body displacement using an inertial measurement unit. With a motion capture system as a reference, the results demonstrated that ankle-based IMU sensors with a tracker system had the potential to replace video-based gait study methods. This collection of literature suggests most studies conducted to date use either accelerometer or gyroscope units in isolation or focus on the different anatomical placements of the IMU. However, the accuracy of gyroscope and accelerometer-based algorithms with IMU placement at the ankle requires further exploration. Furthermore, despite the prospect of enhancing precision, there is an absence of research proposing a system to detect IC and TC using both gyroscope and accelerometer data to improve measurement precision.

Therefore, this study aimed to develop adaptive algorithms based on gyroscope and accelerometer data for the detection of IC and TC during running. The 95% of confidence interval was used to analyse the consistency of the system by examining whether the difference between the output of the two algorithms exceeded the highest tolerance value. Furthermore, to increase the estimation accuracy, we included a pre-processing algorithm that detected the period of running to remove unwanted data such as standing still or mounting the IMU.

## 2. Materials and Methods

### 2.1. The Inertial Measurement System

The inertial measurement unit (IMU) (Blue Thunder, I Measure U, New Zealand, weight: 12 g, size: 40 × 28 × 15 mm) was used in the data collection protocol. The units used a sampling rate of 500 Hz and measured 3-dimensional acceleration and 3-dimensional angular velocity. The IMU system had a detection range of ±16 g for the accelerometer and ±2000°/s for the gyroscope. The collected IMU data were analysed using customised MATLAB (version 2019a, The Mathworks Inc., Natick, MA, USA).

### 2.2. Participants

Two sets of data were used for the two respective parts of this study—referred to herein as the development dataset and the validation dataset. The development dataset was collected from 36 players from an Australian Football League team who were all healthy and did not suffer from any symptomatic musculoskeletal injuries. During the protocol, the participants undertook 30 m running efforts at an estimated 75%, 85%, and 95% of maximum speed, with two trials for each speed, as part of normal training. This 30 m running effort was immediately preceded by a run-in where each participant developed their speed to the target. Each effort was separated with a self-determined recovery period of passive standing. During running, all participants wore their own running shoes. The IMU device was attached to the ankle (immediately superior to the lateral malleolus) of each foot of the participant (represented in Figure 1) with a fixed and standardised orientation. Overall, 432 sets of data right and left ankle unilateral IMU data, 36 sets with 3 running speeds and 2 trials each) consisting of 53,280 steps were collected which were used for algorithm building.

For the validation dataset collection, two healthy participants (without musculoskeletal injury) ran at three different speeds while wearing the IMU device attached immediately superior to the lateral malleolus on each limb as outlined above. Participants undertook a 10 m run through at 3 self-selected speeds (slow, medium, and fast), with each speed repeated for five trials and separated by a self-determined recovery period of passive standing. The data collection was carried out in an enclosed laboratory setting while being filmed at a close distance by a camera system at 30 FPS (iPad, Apple Inc., Sydney, Australia). During running, the participants were wearing their own comfortable shoes. In this study, only the data collected from gyroscope and the accelerometer was used. Overall, 60 trials (two participants, right and left unilateral ankle IMU data, three different speeds and five trials each) involving 180 steps were collected, and these data were used for consistency and accuracy validation.

### 2.3. Data Pre-Processing Development

Following IMU data collection, the raw data contained noise and data from untargeted (non-running) periods such as mounting the IMU, walking or standing still. These sections of the data are removed for the running stance phase detection. Hence, a pre-processing algorithm designed to detect the running period was introduced. This algorithm first calculated the square value of the acceleration in the three directions as shown in Equation (Equation 1). The square value first rectified the acceleration signal into positive and then sufficiently amplified the difference between non-running period and running period. This algorithm contained a sliding window of 0.5 s and compared the average square value of the sum of accelerations in three directions to a threshold value within the window. If the value output exceeded the threshold for more than 3 s, it was marked as the beginning of running, and when the value dropped below the threshold, it was marked as the completion of running. The calculation of the sliding average value is shown in Equation (Equation 2).
(1)$$A_{res}=\left(Ax^{2}+Ay^{2}+Az^{2}\right)$$
(2)$$A_{threshold}=\frac{\sum_{n}^{n+s*f}A_{res}}{s*f}$$
where $A_{threshold}$ is the output value that is compared to the threshold. *f* is the sample frequency of the IMU. $A_{x},A_{y},A_{z}$ are the linear acceleration value in three directions. *S* is the time frame of the sliding window.

### 2.4. Algorithm Design

Based on the data collected from the gyroscope and accelerometer, two different algorithms were created for IC and TC detection using the development dataset.

#### 2.4.1. Accelerometer-Based Algorithm

In the accelerometer-based IC and TC detection, the algorithm was built using acceleration data from the *z*-axis (vertical). Before plotting the information, the square value of acceleration was computed to obtain the positive magnitude and to amplify the data. The instant of IC was detected based on the peak of foot resultant acceleration (Figure 2). The detection of the TC was conducted in the region of interest, which was defined within the 25% to 75% range of a full stride (IC to IC), and when the fluctuation of the *z*-axis acceleration started to show an upward trend after the peak point (IC). The area of interest commenced when the magnitude of the acceleration started to show an upward trend which exceed 2 g and terminated when the signal finished a downward trend and dropped below 2 g. Figure 2 demonstrates the TC point and the area of interest for the TC detection.

During running, as the speed increases, the ground contact time of the runner tends to decrease, hence the ground contact time is often less than 50% of the total stride time during high-speed running [18]. Therefore, to find the exact point of TC, we designed a window to highlight the area of TC detection. This window has a length of N samples, which is half of the total number of samples in a specific stride, and the window is located between 25% and 75% of the stride. Equation (Equation 3) shows the calculation of N. Following this detection, all the data within this region was normalized using Min-max normalization and scanned until the first point that exceeded the empirical threshold of 2 g which was conceivably due to the high inertial change due to the foot leaving the ground. That point was marked as the TC.
(3)$$N_{window}=\frac{1}{2}*\left(Peak_{n+1}-Peak_{n}\right)$$

The pseudo-code of the acceleration algorithm on terminal contact points detection is shown in Algorithm 1.
**Algorithm 1:** Finding TC using Acc data.     **Output:**
TC1: StartTime←fromPre−processing2: EndTime←fromPre−processing3:4: AccZ←AccelerationZ(FromStartTimetoEndTime)5: AccY←AccelerationY(FromStartTimetoEndTime)6: AccX←AccelerationX(FromStartTimetoEndTime)7:8: $resAcc\leftarrow \sqrt{AccX^{2}+AccY^{2}+AccZ^{2}}$9: SqreDiffAcc←((resAcc(2)−resAcc(1)), (resAcc(3)−resAcc(2)),……${\left(resAcc\left(n\right)-resAcc\left(n-1\right)\right))}^{2}$10: [PeakValue,InitialContactx]←localmaximum(SqreDifffAcc)11:12: Flag←013: Count1←114: Count2←115: Peakx←016:17: **for**
i←0toLength(SqreDiffAcc)
**do**18:     **if** Flag=0,SqreDifffAcc(i)=PeakValue(Count1) **then**19:         Flag←120:         Peakx←i21:         INCREMENTCount122:     **else**23:         **if** Flag=1,i=(Peakx+(InitialContactx(Count1)−Peakx)/4) **then**24:            **for** h←ito(Peakx+(InitialContactx(Count1)−Peakx)/2) **do**25:                NrmlsACC←(SqreDiffAcc(h)−Minimum(SqreDiffAcc(hto(Peakx+(InitialContactx(Count1)−Peakx)/2))))/(Maximum(SqreDiffAcc(hto(Peakx+(InitialContactx(Count1)−Peakx)/))))−Minimum(SqreDiffAcc(hto(Peakx+(InitialContactx(Count1)−Peakx)/2))))26:                **if** NrmlsAcc≥EmpiricalThreasholdAcc **then**27:                    TC=h28:                **end if**29:            **end for**30:            TC(Count2)←TC31:            INCREMENTCount232:            Flag←033:         **end if**34:     **end if**35: **end for**

#### 2.4.2. Gyroscope-Based Algorithm

Separate from the accelerometer-based detection in the gyroscope-based algorithm, detection of mid-swing (MS) is integral to determine presence and order of IC and TC. These create unique signal characteristics whereby angular velocity data show distinctly positive or negative peaks consisting of medium to high frequencies [19]. The instance of IC and TC are defined at the first and the second local maximum, where MS is the local minimum that is located after TC, the locations of IC, TC and MS are shown in Figure 3. While the scale of such peaks are affected by several factors such as the participant’s running intensity or speed, they tend to show similar trends among different runs and hence can be detected in the specific area of interest or frequency domains. Based on previous work on gait analysis by Lee and Park [20], who proposed a detection algorithm by first detecting the location of mid-swing (MS), we replicated this technique by subsequently searching backwards for TC and forwards for IC.

The proposed algorithm first detected TC to minimise the computational requirements for backward searching. To enhance and detect the mid-swing peak, a 2nd-order Butterworth low-pass filter was designed with a cut-off frequency of 10 Hz. In the detection of the possible location of TC, the proposed system initially detected the first local maximum point and subsequently searched for the presence of a local minimum point after TC which was the MS. Finally, the following local maximum was then recognised as the location of IC. These possible points of gait event location were then evaluated under the set of rules shown in Table 1.

### 2.5. System Validation

This section outlines the methods used to examine the consistency and accuracy of the two algorithms and was carried out upon the validation dataset. A previous systematic review [21] reported use of 95% confidence intervals to compare the results of gait parameters in studies on patients with inflammatory arthritis. In this study, to test the consistency of the output of gyroscope and accelerometer algorithms, the 95% confidence interval (CI) and 2-sample *t*-test were used to compare and verify the range of the difference between the two algorithms. In the case of a CI in a particular measure *X*, the sample size is defined as *N*, where *m* is defined as the hypothetical average value or mean, and sd is the standard deviation. The pre-defined confidence level is represented as 1−α and in this study a 95% confidence level was selected. Hence, the CI of the given sample or the range of values can be defined by $m-c_{\alpha }s<X<m+c_{\alpha }s$, and as α=0.05, then cα=1.96. To test the consistency between the two algorithms, a right-side test was conducted to the difference between the calculated contact time of the two algorithms $(H_{0}:\mu \leq \mu_{0}\;|\;H_{1}:\mu >\mu_{0},\mu_{0}=0.03\text{–}0.05$), where $H_{0}$ is the null hypothesis and $\mu_{0}$ is the acceptable tolerance range and μ is the mean value of the algorithm result. The average of the absolute difference between the two algorithms ${\bar{x}}_{diff}$ is calculated by Equation (Equation 4). The z-score value is then calculated using Equation (Equation 5) and the output *Z* value is then compared to the Critical Value Zα/2 which is 1.654.
(4)$${\bar{x}}_{diff}=\frac{\sum_{i=1}^{n}\left|CT_{Gyro}-CT_{Acc}\right|}{n}$$
where ${\bar{x}}_{diff}$ is the average value of the absolute difference between the outputs from the two algorithms. $CT_{Gyro}$ and $CT_{Acc}$ are the calculated contact time output from the two algorithms.
(5)$$z=\frac{{\bar{x}}_{diff}-\mu_{0}}{sd/\sqrt{n}}$$
where *z* is the z-score value of the current ${\bar{x}}_{diff}$, $\mu_{0}$ is the maximum tolerance of the ${\bar{x}}_{diff}$ value, and sd is the standard deviation of the ${\bar{x}}_{diff}$ dataset.

The output z-score value is then compared to assess whether the system’s consistency is within the confidence level of 95% where the maximum inconsistency of the two algorithms is 0.03 s due to the frame rate per second of the video ground truth.

The accuracy of the system was validated using the video footage of the gait parameters. Specifically, this was undertaken by manually recording the actual instance of IC and TC of each step and calculating the CT from the validation dataset, with ab-solute error calculated and recorded. The outputted IC, TC, and CT from the two system were then compared using the 2-sample *t*-test, the percentage of the unrejected hypothesis tests was recorded as the systems’ accuracy. Results are presented as absolute mean ± standard deviation (sd). Significance was set at α = 0.05. Using the Shapiro–Wilk test, visual analysis of the histograms and box plots, the normal distribution of the data were confirmed.

## 3. Results

### 3.1. Data Pre-Processing

The purpose of the pre-processing algorithm was to filter out the non-running data ranges and to output the running period start time and end time for IC and TC detection in the later part of the algorithm. Figure 4 demonstrates a random sample of a participants’ right-foot data from the validation data set, including all three different run through efforts. The algorithm clearly distinguished the three different running periods from the whole data set (519 s in duration). Furthermore, the different speed 10 m efforts were clearly demonstrated with three distinct ranges of squared resultant acceleration peak values. As the sliding window has a length of 0.5 s and the algorithm used average values within the window, the output value could contain minor errors of up to 0.5 s which could affect the detection of IC and TC on a specific step. However, one step can be considered as a minor influence on the algorithm’s accuracy given the large number of steps in the data set.

### 3.2. Algorithm Consistency

To test the consistency of the gyroscope and accelerometer algorithms, 95% right-tailed confidence intervals with different maximum tolerance value (0.05 s, 0.04 s and 0.03 s) were constructed on the difference between the output value of the two algorithms. Table 2 summarizes the consistency of the two algorithms at different speeds. The consistency was computed by calculating the percentage of datasets with a lower difference than the given tolerance (<95% confidence level) between the output of the two algorithms. As depicted in Table 2, a 100% consistency between the two systems existed for all speeds when a maximum difference value of 0.05 s was used. An average consistency of 94.44% was evident for CT in all datasets when using a maximum difference of 0.04 s and for the lowest level of tolerance (0.03 s), an average consistency of 86.67% was demonstrated. The two algorithms showed the highest consistency at a medium speed, and the average consistency was 94.63%. From the results, we found that the consistency of the two algorithms reduced as the acceleration speed increased to fast. In addition, the highest consistency for each tolerance group was evident at the medium acceleration speed.

### 3.3. Accuracy of Detection

After the consistency of the two algorithms was assessed, the output values of the two algorithms were then compared to the video reference to test the validity of the systems. Table 3 shows the mean absolute errors (MAE) of the accelerometer algorithm and gyroscope algorithms. By comparing the accuracy of the two algorithms at three different running speeds, we found that at the medium speed, both algorithms showed higher consistency and accuracy and the mean errors were at their lowest, 0.0273 s and 0.0214 s. Another interesting trend is that both the accuracy and consistency were at their lowest during high-speed acceleration as the average consistency was at its lowest of 91.3% and the average mean error of both the acceleration and angular velocity data were at their highest of 0.0364 s and 0.029 s. When compared to other studies, the common MAE of IC and TC detection ranged between 10 and 60 ms [2,4,22,23,24], while our study yielded an MAE between 5 and 37 ms. Overall, these results show that both the accelerometer and gyroscope have potential in developing IMU-based gait analysis algorithms for CT and FT detection.

Given the observed reduction of both consistency and accuracy of the two systems at higher speed, we performed a preliminary examination of the ground truth video for the algorithm performance based on foot strike type. Rear-foot strike means the heel of the runner makes the first contact with the ground whereas fore-foot strike is when the heads of the metatarsals of the runner contacts the ground first [25]. Table 4 summarizes the consistency of the two algorithms under different speeds and strike types. Given the increased presence of fore-foot strike at higher speeds, the fore-foot strike could affect the consistency and accuracy of the system. As observed in Table 4 the two systems showed relatively low average consistency under fore-foot strikes, whereas the systems’ consistency remained high under rear-foot strikes regardless of the speed.

## 4. Discussion

This study aimed to develop and validate adaptive algorithms based on gyroscope and accelerometer data for the detection of IC and TC during running from IMU’s mounted on the ankle. The proposed algorithm successfully detected the running period and removed the noise with the pre-processing system. Furthermore, results showed high accuracy and consistency during in-field running with slow to fast running speeds. Therefore, the algorithm we proposed can successfully identify the key gait events of IC and TC in many running-related applications during in-field settings. Such outcomes can overcome the limitations related to other laboratory-based gait analysis methods (motion tracking system or camera-based systems), which are difficult to use in field-based settings. Moreover, such a system is desirable as it provides the potential of analysis and feedback of running gait parameters for athletes and coaches during field-based training or competition. In support, Muro et al. [15] conducted a review of current gait analysis methods, reporting that the high level of performance of IMU-based algorithms in terms of accuracy, accessibility, and transportability made them more suitable than other gait analysis approaches during athletic applications. With these advantages, the application of accelerometer and gyroscope-based algorithms for IC and TC detection may present an avenue for further research for healthy athletes [22], athletes with transtibial amputation [26] or athlete recovery [27].

The current study showed that the gyroscope-based algorithm can maintain the same waveform shape under different or increasing running speeds as only the magnitude of the angular rate and frequency of the foot are affected. For example, the current study showed that the absolute mean error presented less variation when running speed increased when compared to the acceleration-based algorithm. Furthermore, our results demonstrated that the gyroscope algorithm had similar mean absolute error (MAE) for IC and TC (11.3 ms and 16.7 ms, respectively). Similar observations have been previously reported [23] whereby the running gait of healthy and spinal cord injured individuals were examined. Their results revealed a MAE of 12 ms for IC and 15 ms for TC in normal groups, whereas the spinal cord injured groups revealed MAE of 20 ms in IC detection and 22 ms in TC detection. Despite these advantages, gyroscope-based algorithms usually require wavelet transformation and different filter design according to different speeds, running intensity and IMU placement. For example, Gouwanda et al. [22] designed a 2nd order digital low pass filter with cut-off frequency of 15 Hz to highlight the waveform peaks for IC and TC detection. McGrath et al. [24] used a low pass filter with zero-phase fifth order Butterworth filter with a 50.2 Hz corner frequency.Conversely, the cut-off frequency was set to 10 Hz to obtain the best waveform information in the current study. It is noteworthy that the performance of these algorithms can be affected with expanded sample size or different running styles, and future work in this area is required to refine algorithm accuracy.

For the accelerometer-based algorithm, the detection of TC is critical and usually generates more errors compared to the detection of IC (Jasiewicz et al., 2016). Conversely, IC detection is often highly accurate and adaptive to different running speed due to the high velocity change when the foot strikes the running surface. The current results demonstrated this trend by yielding higher errors in TC than IC, as the MAE of TC ranged from 23.8 ms to 31.4 ms, while IC was 4.5 ms to 6.7 ms. In support, Jasiewicz et al. [23] studied accelerometer-based gait study algorithms with IMU placement on the foot and suggested that the MAE of TC (27.6 ms) was substantially higher than the MAE of IC detection (4.2 ms). Moreover, the accuracy of the TC detection in acceleration-based analysis has not been widely investigated. Accordingly, future research should investigate the accuracy of TC detection and the factors which lead to the reduction in accuracy.

The combination of using both accelerometer and gyroscope-based algorithms appears to be a sound approach to mitigate the limitations of these two respective gait analysis methods. In this study, we investigated the consistency of the two algorithms with 95% confidence interval and discovered that the proposed systems demonstrated good overall consistency, despite reduced consistency (81.67%) for TC detection at higher running speeds. A similar tendency was presented by Piriyakulkit et al. [28] where they proposed a gait event recognition algorithm to assist patients with lumbar kyphosis using IMU. The proposed algorithm recorded MAE’s of 31.6 ms, 32.4 ms, and 38.6 ms in TC detection under increasing treadmill speeds. In our study, the two algorithms showed relatively low average consistency for fore-foot strikes, whereas the systems’ consistency remained high under rear-foot strikes regardless of the speed. Hence, we suspect that this change in the detection accuracy is possibly due to the foot strike style (fore-foot or rear-foot) and requires further investigation and refinement. Accordingly, this study shows the feasibility of using the combination of both algorithms, where the advantage of each approach could be used to further improve the detection accuracy of IC and TC. An example of this approach could be the use of a weighted algorithm in which the accelerometer-based system has a larger influence in IC calculation and the gyroscope-based system being more impactful in TC detection. Accordingly, future studies should investigate the effect of different striking styles have on the current gait algorithms alongside the accuracy of TC detection based on the combination of accelerometer and gyroscope-based systems with increased experimental sample sizes and variety of foot-striking characteristics, running speeds or IMU placement locations.

## 5. Conclusions

In this paper, we proposed an intelligent running gait analysis system that can estimate contact time and flight time by detecting key running instances such as IC and TC with high accuracy and consistency. The system used two algorithms which were designed based on the data from the gyroscope and accelerometer. We also introduced a pre-processing algorithm to detect the running period to enhance the detection accuracy. Furthermore, the consistency of the two algorithms was studied using a 95% confidence interval and the accuracy of the system was investigated using a validation data set. The results showed that the accelerometer and gyroscope combined system can obtain the desired accuracy. Our ultimate objective is to design a highly accurate IMU-based gait analysis system which combines the information obtained from the gyroscope and accelerometer. The system should be compatible to both striking styles (rear-foot and fore-foot) that may affect the systems’ consistency and accuracy in the proposed algorithms. Therefore, future research focussing on the effect of the striking type on the current gait algorithms and different IMU placement locations could further improve the compatibility and detection accuracy of the system.

## Figures and Tables

**Figure 1 sensors-22-04812-f001:**
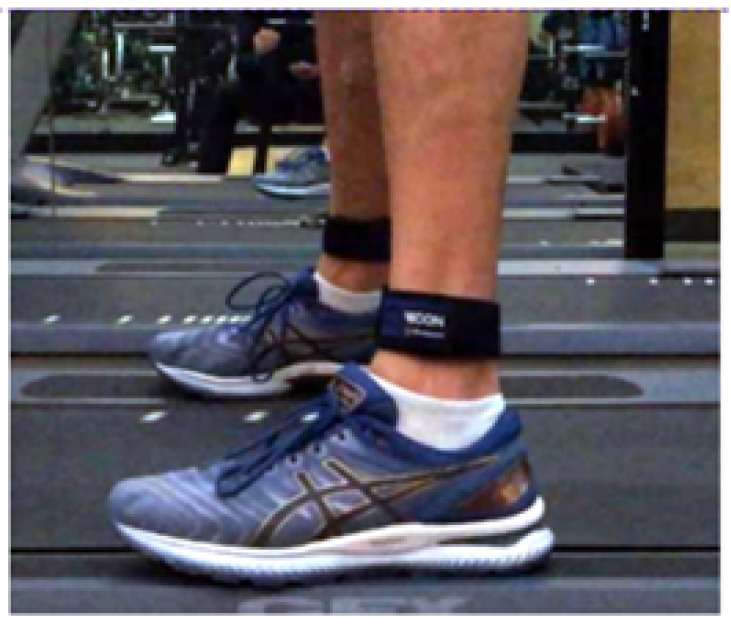
Example placement of the IMU device on the ankle.

**Figure 2 sensors-22-04812-f002:**
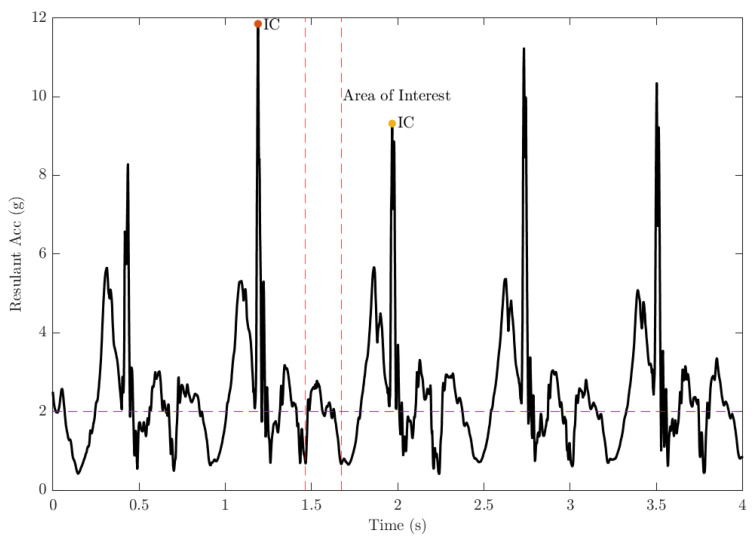
Random samples of right ankle acceleration value in g $A_{ankleR}$ of a participant during 85% maximum speed running. Please note that the peak resultant acceleration is marked as IC, and the 2 g-threshold is the area of interest for TC detection.

**Figure 3 sensors-22-04812-f003:**
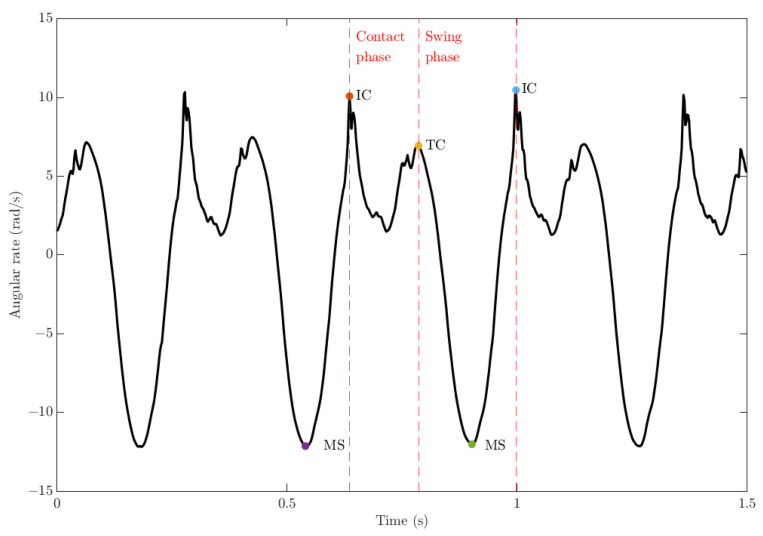
Random samples of right ankle angular rate $\omega_{ankleR}$ of a participant during 75% maximum running speed. Please note that MS = Mid-Swing; IC = Initial-Contact; TC = Terminal-Contact.

**Figure 4 sensors-22-04812-f004:**
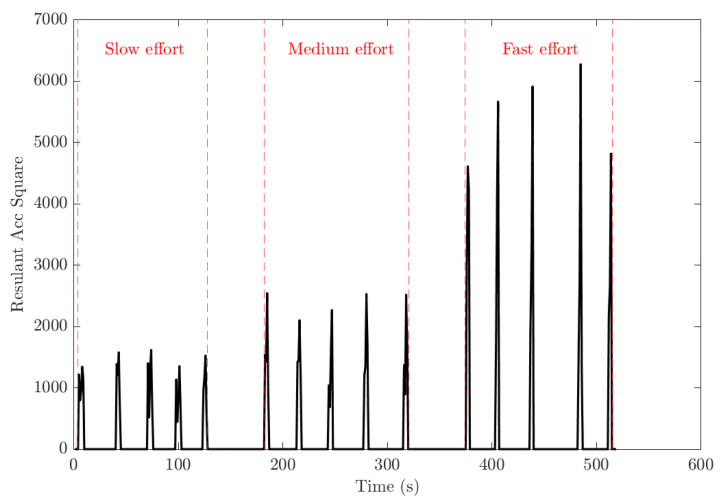
Random sample of a participant’s pre-processed right-foot data with three different 10 m run through efforts (slow, medium, fast).

**Table 1 sensors-22-04812-t001:** The detection logic and conditions for IC, MS and TC detection.

Gait Event	Conditions
**TC**	TC must fulfil below conditions:
	(a) It is the local maximum
	(b) A local minimum MS is after $TC_{pp}$
	(c) A local maximum IC is after MS
	(d) It is the local maximum between $MS_{n}$ and $MS_{n-1}$
**MS**	*MS* must fulfil below conditions:
	(a) It is the local minimum
	(b) MS(n,1)–MS(n−1,1)> 300 ms
**IC**	IC must fulfil below conditions:
	(a) A MS is identified before the location of IC
	(b) IC(n,1)–TC(n,1)> 100 ms

**Table 2 sensors-22-04812-t002:** The consistency of the two systems under different speeds and maximum tolerance values of the confidence interval.

Percentage of Datasets within Maximum Tolerance Value with 95% CI (s)										
Speed	0.05			0.04			0.03			
	CT	IC	TC	CT	IC	TC	CT	IC	TC	Average
Slow	100%	98.33%	100%	95%	96.67%	93.33%	86.67%	85%	86.67%	93.52%
Medium	100%	100%	100%	96.67%	96.67%	95%	88.33%	90%	85%	94.63%
Fast	100%	96.67%	98.33%	91.67%	91.67%	90%	85%	86.67%	81.67%	91.3%
Average	100%	98.33%	99.44%	94.44%	95%	92.78%	86.67%	87.22%	84.44%	

**Table 3 sensors-22-04812-t003:** The mean error of the two algorithms under different speeds.

Speed	Absolute Mean Error (ms)					
	Accelerator Algorithm			Gyroscope Algorithm		
	IC	TC	CT	IC	TC	CT
slow	5.8 ± 2.1	27.5 ± 11.7	30.9 ± 12.1	12.1 ± 6.7	15.1 ± 5	23.9 ± 10.3
medium	4.5 ± 2.3	23.8 ± 9.3	27.3 ± 13.4	9 ± 10.7	13.7± 7.8	21.4 ± 15.9
fast	6.7 ± 3.2	31.4 ± 14.6	36.4 ± 16.1	12.8 ± 9.3	21.4 ± 8.5	29 ± 10.4

**Table 4 sensors-22-04812-t004:** The consistency of the two systems under different speed and maximum tolerance value of the confidence interval.

	Fore-Foot Strike			Rear-Foot Strike		
**Speed**	**Appearance**	**Average** **Consistency**	**CT** **MAE (ms)**	**Appearance**	**Average** **Consistency**	**CT** **MAE (ms)**
Slow	0%	N.A	N.A	100%	100%	27.4 ± 11.5
Medium	8.75%	42.15%	25.5 ± 13.8	91.25%	96.25%	24.3 ± 15.7
Fast	86.25%	35.75%	33.3 ± 14.5	13.75%	91.25%	29 ± 11.4

## Data Availability

The data that support the findings of this study are available from the corresponding author upon request.

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
