# Peer review of "Inertial Sensor Estimation of Initial and Terminal Contact during In-Field Running"

_sensors, 2022, doi:10.3390/s22134812_

Round 1

Reviewer 1 Report

This study presents results from estimation of contact time and flight time during running at different speeds using inertial sensors (accelerometers and gyroscopes). 

Overall, the study lacks novelty. The algorithms proposed for event detection appear to be heuristic and anecdotal, based on visual inspection of the temporal waveforms. 

1. The title suggests that the study is about estimation of FT and CT. However, in the Introduction section, detection of IC and TC is described as main aim of the study.

2. The authors state in Introduction "However, the accuracy of using gyroscope and accelerometer data at the same location is yet to be reported." This may be an overstatement. The authors are advised to look at  https://doi.org/10.1038/s41598-021-81009-w and the references there in.

3. The authors used a sliding window of 0.5 sec for detection of running. In fast running at 165 paces per minute this window size may include up to three events of interest. The authors acknowledge this source of error on pg.8. A smaller window size (0.1s) should perhaps be used in the study.

4. In accelerometer placement the orientation of the sensors is critical for data collection. The authors seem to have neglected this aspect.

5. When comparing the value to a threshold, both quantities should have the same units. The authors seem to neglect this aspect in (1) and (2).

6. The detection of running based on (1) and (2) appears to be anecdotal. The authors may take help from Fourier analysis to separate stationary and running periods.

7. Sec. 2.5 uses 2 sample t-test to compare the results of the two algorithms only for CT. The hypothesis testing should be performed for IC and TC as well.

8. How was acceptable tolerance range (0.03-0.05) determined? How about using 0.01-0.02s?

9. What is an acceptable value of MAE for validation (Table 3)? How is it determined?

10. The consistency of the two sensors for fast running is low (36%). This calls for an improvement in the heuristic detection algorithms. Also, adverse consistency should be reported in the Abstract. 

11. The reliability claim in Discussion (pg. 10, line 242) is not supported by evidence.

12. As suggested in Discussion, the data from both sensors should be combined to improve reliability. This should be part of the study.

Reviewer 2 Report

This paper proposed an intelligent analysis system from wireless micro–Inertial Measurement Unit (IMU) data to estimate contact time (CT) and flight time (FT) during running based on gyroscope and accelerometer sensors in a single location (ankle).A pre-processing system that detected the running period was introduced to analyse and enhance CT and FT detection accuracy and reduce noise. Results showed pre-processing successfully detected the designated running periods to remove noise of non-running periods.

The work is interesting and I am keen to see it published; however, there are a few changes that could be made to make the results more accessible and clear to readers, in details:

1.The motivations should be strengthened when revising in the introduction. The authors should further enlarge Introduction with current results to improve the research background.

2.Transitions from section to section should be smoother

3.Table 4 is not quite informative.

4.More literature may be provided in the experimental results section as references. Some examples are:

- Marta, G., Alessandra, P., Simona, F., Andrea, C., Dario, B., Stefano, S., Stefano, M.. Wearable Biofeedback Suit to Promote and Monitor Aquatic Exercises: A Feasibility Study. IEEE Transactions on Instrumentation and Measurement, 2020, 69(4), 1219-1231.

- S. Qiu, H. Zhao, N. Jiang, D. Wu, G. Song, H. Zhao, Z. Wang. Sensor network oriented human motion capture via wearable intelligent system, International Journal of Intelligent Systems, 2022, 37(2): 1646-1673.

5.What are the implications of the findings? More discussion should be provided in the manuscript. It is not enough to have a paper accepted by just reporting what they have done.

6.Some individuals are not willing to wear additional sensors on their body, what is the potential of smartphone serving as the data collection tool?

7. Proofread the paper and improve readability.

Round 2

Reviewer 1 Report

Most of the previous concerns were addressed in the revised version.

Reviewer 2 Report

The Revised Paper has incorporated all the necessary suggestions as mentioned in the last review and now the paper stands Accepted with no further revisions.